# Factors Influencing Hospitalization Rates and Inpatient Cost of Patients with Tuberculosis in Jiangsu Province, China: An Uncontrolled before and after Study

**DOI:** 10.3390/ijerph16152750

**Published:** 2019-08-01

**Authors:** Dan Hu, Qian Long, Jiaying Chen, Xuanxuan Wang, Fei Huang, John S. Ji

**Affiliations:** 1School of Health Policy & Management, Nanjing Medical University, Nanjing 211166, China; 2Creative Health Policy Research Group, Nanjing Medical University, Nanjing 211166, China; 3Global Health Research Center, Duke Kunshan University, Jiangsu 215316, China; 4National Center for Tuberculosis Control and Prevention, China CDC, Beijing 102206, China; 5Environmental Research Center, Duke Kunshan University, Jiangsu 215316, China

**Keywords:** Tuberculosis, hospital admission rates, health financing and payment, financial burden, China

## Abstract

Objective: The China Center for Disease Control and Prevention (CDC) introduced an innovative financing model of tuberculosis (TB) care and control with the aim of standardizing TB treatment and reducing the financial burden associated with patients with TB. This is a study of the pilot implementation of new financing mechanism in Zhenjiang, between 2014–2015. We compared TB hospitalization rates and inpatient service costs before and after implementation to examine the factors associated with hospital admissions. Our goal is to provide evidence-based recommendations for improving TB service provision and cost control. Methods: We reviewed new policy documents on TB financing. We conducted a patient survey to investigate the utilization of inpatient services, and patients’ out-of-pocket payment for inpatient care. We extracted total medical expenditures of inpatient services from inpatient records of TB designated hospitals. Findings: 63.6% (*n* = 159) of the surveyed patients with TB were admitted for treatment in 2015, which was higher than that in 2013 (54.8%, *n* = 144). The number of hospital admission was slightly lower in 2015 (1.16 per patient) than in 2013 (1.26 per patient), while the length of hospital stay was longer in 2015 (24 days) than in 2013 (16 days). In 2015, patients from families with low incomes were more likely to be admitted than those from higher income groups (OR = 3.06, 95% CI: 1.12–8.33). The average inpatient service cost in 2015 (3345 USD) was 1.7 times the cost in 2013 (1952 USD). It was found that 96.2% of patients with TB who were from low-income households spent more than 20% of their household income on inpatient care in 2013, versus 100% in 2015. Conclusion: The TB hospital admission rate and total inpatient service cost increased over the study period. The majority of patients with TB, particularly poor patient who used inpatient care, continue to suffer from heavy financial burden.

## 1. Introduction 

Tuberculosis (TB) is a chronic communicable disease. Patients with TB admitted to healthcare facilities may reduce the risk of infection among their household contacts. However, the risk of infection for patients treated at home was found to be no greater than inpatients, as transmission primarily occurs before diagnosis [1,2]. International recommendations suggest 6 months of outpatient treatment for patients with TB without complications [3], and inpatient services for those who are homeless or living in a group setting, with severe complications, comorbidities, and adverse reactions [4,5,6]. Hospitalization has several negative consequences: increased risk of transmission to health care providers and other vulnerable patients, separation from social support systems, feelings of isolation, and lost income [7]. Furthermore, several studies have found that hospitalization accounts for more than half of total treatment costs [5,8,9]. An Australian study estimated that by reducing the proportion of TB inpatients from 55% to 45%, the total cost of treatment was reduced by nearly 28% [5].

China is a TB high burden country, with 889,000 new cases in 2017 [10]. Although the government issued a “free policy” for TB diagnosis and treatment, paying for essential X-ray and sputum tests and first-line anti-TB drugs by public funds, previous studies found that over-treatment and over-prescription were common [11,12]. China’s CDC (Center for Disease Control) conducted an institutional survey in 2010 and found that the hospital admission rate of patients with TB increased by 185.3% between 1999 and 2009, with an associated increase of inpatient service costs [13].

Since the 1990s, the national TB control program required that anyone suspected of carrying TB should be referred to TB dispensaries or the TB department in prefectural and county/district CDCs, which were in charge of TB diagnosis, treatment, and case management applying the World Health Organization (WHO) recommended Directly Observed Treatment, Short-course (DOTS) strategy. TB patients with complications were referred to general hospitals for treatment [14]. Recently, China’s National 12th Five-year Plan for the Tuberculosis Control proposed to integrate TB care into general hospitals to provide TB diagnosis and treatment and cooperate with primary health facilities and local CDC for case management. This is called the “designated hospital model” for TB control. The previous study found that TB hospital admission rates and inpatient costs were high in designated hospitals, which was partly associated with health professionals’ profit-driven behavior, and partly due to failures of health insurance regulations [15]. 

With the support from the Bill and Melinda Gates Foundation, China’s CDC introduced an innovative financing and payment mechanism for TB care aiming to standardize TB treatment based on treatment clinical pathways and reducing the financial burden on patients with TB (called “China-Gates TB Project Phase II”). This was piloted in three prefectural cities in 2014–2015. Zhenjiang, a city in Jiangsu province in the eastern region of China, was one of the project sites. This study compared the TB hospital admission rates and inpatient service costs before and after the implementation of a new financing mechanism in Zhenjiang, and examined the factors associated with hospital admissions and costs to provide evidence-based recommendations to improve TB service provisions and cost containment.

## 2. Materials and Methods 

Three counties in Zhenjiang, Dantu, Yangzhong, and Jurong, were selected and included in the pre- and post-intervention investigations. Multiple data sources were used, and are listed below. 

### 2.1. Data Sources

#### 2.1.1. Policy Review

We reviewed local health insurance regulations and reimbursement policies at the county and municipal levels before the introduction of the new financing mechanism and issued new policies related to financing and payment method during the project period. 

#### 2.1.2. Patient Survey

Cross-sectional TB patient surveys were conducted at pre-intervention in 2013 and post-intervention in 2015. We selected three townships in each county, based on the distance to the county TB designated hospital (far, not far, and close) with the first 90 patients sampled from the list of TB patient registration for both surveys. The inclusion criterion at the baseline was that patients had completed TB treatment before the survey; at the post-intervention survey, the criterion was that patients started and completed treatment during the intervention period. The questionnaire included questions on the general demographic and socio-economic background of the sampled patients, pathways of seeking TB care, and utilization of services and out-of-pocket payment. The sampled patients were all invited to the township health centers by village doctors; only a small number of patients refused. Trained medical students from Nanjing Medical University conducted face-to-face interviews. There were 263 patients in 2013 and 250 in 2015 in the analysis after data cleaning. 

#### 2.1.3. Inpatients Records 

We collected all inpatient records of patients with TB from the city and county designated hospitals between 2013 and 2015. The information included diagnoses at dates of admission and discharge, length of hospital stay, content of care, and costs. We linked inpatient records with a patient survey database of those admitted using patients’ names. Out of the 159 admitted patients in the patient survey, 118 cases were successfully linked to hospital inpatient records. The possible reason for unlinked cases (*n* = 41) was that they may have been admitted in non-designated hospitals. 

#### 2.1.4. Data Analysis

We defined hospital admission by the use of inpatient medical services for more than one day. Based on the patient survey, the main outcomes included hospital admission rate (defined as the number of admitted patients with TB/total number of patients with TB), the average number of hospital admissions per patient, the length of the hospital stay for each admission, and self-reported out-of-pocket payment for inpatient care. The total out-of-pocket payment for inpatient care was a sum of patients’ self-reported direct medical costs for each hospital admission as a result of TB. We used the out-of-pocket payment as the main indicator to assess the financial burden on households. 

Also, length of hospital stays and total medical cost for inpatient care (including hospital fees, medication, relevant examinations, and services fees) were examined according to the linked database of patient surveys and inpatient records. The explanatory variables included patient age, sex, household income (which divided into three groups according to the distributions, high income (top 25), middle income (medium 50), and low income (lowest 25)), poverty (those living below local poverty line), type of patients with TB (new patients, relapsed patients), type of health insurance (Urban Employee Basic Medical Insurance, Urban and Rural Residence Basic Medical Insurance), TB bacteriology diagnosis (smear positive, smear negative), and study counties. The medical costs for inpatient care and annual household income were converted into US dollar using Purchasing Power Parities (PPP) in the study years (OECD National Accounts Statistics: PPP, https://data.oecd.org/conversion/purchasing-power-parities-ppp.htm#indicator-chart). 

Data were cleaned and analyzed using IBM SPSS Statistics version 22.0 software (IBM, Armonk, NY, USA). Cross-tabulation was used to compare the utilization of inpatient services, total medical costs, and self-reported out-of-pocket payment for inpatient care before and after the implementation of the intervention. A chi-square test was used to test differences. The association between hospital admissions and the explanatory variables was studied by binary logistic regression analysis. A linear regression model was used to study the correlation between the logarithmic value of total medical costs for inpatient care and explanatory variables. Patients’ self-reported out-of-pocket payment for inpatient care as a percentage of annual household income in 2013 and 2015 was calculated for the three income categories. We also calculated the percentage of patients experiencing catastrophic expenditure, i.e., exceeding 10% and 20% of the household’s annual income, respectively [16,17,18].

#### 2.1.5. Ethical Approval

Ethical approval was issued by the Ethical Review Committee of Chinese Center for Disease Control and Prevention. The approval notice was conducted without ethical approval number. Oral consents of patients with TB who participated in the survey were obtained. Permission to access to hospital inpatient records was obtained from local CDC. 

#### 2.1.6. Patient and Public Involvement

In this study, patients with TB were not involved in the study question development or the study design. 

## 3. Results 

### 3.1. Health Insurance Reimbursement Policy and Payment Method for TB Inpatient Care 

There are two main health insurance schemes in Zhenjiang: urban employee basic medical insurance (UEBMI) and urban and rural residence basic medical insurance (URRBMI). The UEBMI provides relatively comprehensive coverage of health services. The URRBMI, by contrast, is restricted to reimbursements for inpatient services and a limited number of outpatient services. Before the introduction of new financing and payment methods for TB care in TB designated hospitals, health insurance agencies paid hospitals based on a global budget to cap overall healthcare bills. UEBMI provided relatively high reimbursement rates for inpatient care, while the reimbursement rates of URRBMI ranged from 30–75% across three study counties (Table 1). After the implementation of the China-Gates Project Phase II in 2014, Dantu and Yangzhong created a case-based payment approach for TB inpatient care by setting a fixed payment rate for each case at a ceiling amount of 8000 Chinese Yuan (CNY), with the reimbursement rates of both UEBMI and URRBMI being no less than 80%. In Jurong, the payment method for TB inpatient care did not change, but increased the reimbursement rate to 80%. In addition, hospital admission rates should be, at most, 30%.

### 3.2. The Characteristics of Patients with TB from the Patient Survey

Table 2 presents the characteristics of patients with TB in the pre- and post-intervention surveys in 2013 and 2015. In both surveys, more than half of the patients were 60 years old and above, and there were more males than females. The vast majority of patients were enrolled in urban and rural residence basic medical insurance (URRBMI). Around three-quarters of patients were new cases, and most were smear-negative. It was found that 7% and 12% of patients lived under the local poverty line in 2013 and 2015, respectively. 

### 3.3. The Utilization of Inpatient Care and Factors Related to Care Use

The hospital admission rate was 63.6% in 2015, which was higher than that in 2013 (54.8%) (*p* < 0.05) (Table 3). The admission rate among patients from low-income households increased significantly, from 49.1% in 2013 to 73.3% in 2015 (*p* < 0.01). Also, patients who were aged 30–59 and enrolled in URRBMI increased the use of inpatient care between 2013 and 2015. The admission rate increased significantly from 43.3% in 2013 to 68.7% in Jurong county (*p* < 0.01), when the reimbursement rate for hospital admissions increased. In 2015, the hospital admission rate was highest in Dantu, followed by Jurong, and lowest in Yangzhong. In our adjusted analysis, low-income patients were more likely to be admitted (OR: 3.058, 95% CI: 1.123–8.331) after the introduction of the new financing and payment policy (Table 4). Also, study counties were associated with hospital admission rates. The number of hospital admissions per patient was 1.26 per patient in 2013, which was slightly lower (1.16 per patient) in 2015; the difference was not statistically significant. However, the length of hospital stays was longer in 2015 (24 days) compared to 2013 (16 days); again, the difference was not statistically significant.

### 3.4. The Cost of Inpatient TB Services and Factors Related to the Cost 

Based on the inpatient records, the average medical cost of inpatient care in 2015 was 1.6 times that in 2013 (Table 5). Compared to 2013, the average medical cost of inpatient care in 2015 was doubled among the middle-income group, newly-diagnosed patients and smear-positive patients, and was almost tripled among patients covered by UEBMI. The linear regression analysis found that smear-positive patients were associated with higher inpatient costs compared to smear-negative patients in 2015. There were no statistically significant correlations between inpatient costs and other explanatory variables (data not shown). 

Based on patient self-reporting, the out-of-pocket payments for inpatient care as a percentage of annual household income were extremely high among patients from low-income groups in both years. In 2013, the out-of-pocket payments for inpatient care accounted for 100% of annual household income among patients from low-income households, and in 2015, this percentage was 575% (Table 6). It was found that 96.2% of patients with TB who were from low-income households spent more than 10% of their household income on inpatient care on treatment in 2013, and this percentage was 100% in 2015. At the 20% threshold, we found the same results in 2013 and 2015 (Table 6). 

## 4. Discussion

We found that TB hospital admission rates increased, particularly among patients from low-income households and those who were covered by URRBMI after the introduction of new financing and payment methods. A slight increase of admission rates in Dantu and Yangzhong was observed, where the introduced case-based payment for inpatient care and increased the reimbursement rate of basic medical insurances. A significant increase in Jurong was observed, where increased reimbursement rates and a global budget to cap healthcare bills were paid by health insurance agencies. Compared to the baseline in 2013, the medical cost of inpatient care, however, increased substantially after the implementation of the intervention in 2015. The most significant increase in medical costs on inpatient care was found among patients from middle- and high-income groups and those who were covered by UEBMI. The out-of-pocket payment for inpatient care accounted for a very high percentage of annual household income among patients from low-income households. 

Coinciding with the development of health insurance schemes (rural new cooperative medical scheme (NCMS), urban residence basic medical insurance (URBMI), and urban employee basic medical insurance (UEBMI)) in China, the utilization of health services greatly increased in 2003–2011 [19]. According to the 5th National Health Services Survey in China, the hospital admission rate increased by 150% nationally over the period between 2003 and 2013 [20,21]. The increase in hospital admission in rural areas was particularly striking, as the NCMS largely covers inpatient services. In Zhenjiang, NCMS was integrated with URBMI before 2013, called Urban and Rural Residence Basic Medical Insurance (URRBMI), which also provided relatively high reimbursement rates for inpatient care, and had generally limited coverage on outpatient care. The national TB practice guideline recommended that TB treatment largely relies on outpatient care. In this study, we found that more than half of patients with TB were admitted at the baseline survey. This proportion increased far beyond the suggested 30% in the issued new financing and payment policy, and total medical costs for inpatient care almost doubled after the intervention. 

The new financing and payment policy failed to regulate hospital admission rates or contain medical cost, despite the number of hospital admissions per patient being slightly lower after the implementation of the new policy. Possible reasons for this may be hospital profit-driven behavior and the lack of mechanisms to effectively monitor and support policy implementation. TB designated hospitals still largely rely on fee-for-services, and TB care providers’ salaries are linked with the revenue generated for the hospital. It is not surprising, as many other studies in China have reported, that a perverse financial incentive is driving healthcare providers to prescribe more and more expensive health services [22,23,24,25,26]. The increase of total medical costs for inpatient care might be partly attributed to increasing the content care according to TB treatment clinical pathways. However, our another study found that healthcare providers prescribed services beyond the benefit packages based on the case-based payments, and those services were reimbursed according to the original reimbursement policy [27]. Thus, this study found the total medical costs for inpatient care were higher than the defined ceiling amount of 8000 CNY, and that actual reimbursement rates were lower than the suggested reimbursement rate by no less than 80%. Failure in the implementation of the case-based payment method was partly due to the lack of effective administrative sanctions and partly due to invalid supervision by health insurance agencies, given the information asymmetry between health insurance agencies and designated hospitals. Consequently, patients with TB, particularly poor patients who used inpatient care, still suffered from a catastrophic financial burden for medical services. 

This study linked the medical records of admitted patients with TB to the data obtained from the patient surveys in 2013 and 2015. Length of hospital stay and total medical cost of inpatient care are accurate without patient recall bias. However, there are several limitations to this study. Out-of-pocket payment for TB inpatient care was based on patient self-reporting, and may suffer from recall bias. Given that payments for inpatient care were relatively high, serious recall bias is unlikely. In this study, we only examined out-of-pocket payment for direct medical costs associated with TB inpatient care. Direct non-medical costs (e.g., transportation, food and nutrition, accommodation of accompanying members, etc.) and indirect costs (e.g., income loss) due to TB inpatient care are not included. Hence, the estimated financial burden of TB inpatient care is certainly higher. Also, innovative financing models of TB care and control are among the interventions implemented at the project sites. We did not assess the impact of other interventions (e.g., the introduction of new technology in TB diagnosis and treatment, etc.) on the utilization and cost of TB care in this study. Since the sample size was small and geographical coverage was limited in this study, generalization to other areas should be made with caution. 

## 5. Conclusions 

We did not expect to see increased TB hospital admission rates and total medical costs for inpatient care after the introduction of the new financing and payment method, despite the fact that the use of inpatient care increased among patients from low-income households. The majority of patients with TB, particularly poor patients who used inpatient care, still suffered from a heavy financial burden. Implementation research to explore effective approaches to improving the quality of care and containing medical costs through the engagement of stakeholders from multiple institutions and agencies are urgently needed. 

## Figures and Tables

**Table 1 ijerph-16-02750-t001:** Health insurance reimbursement policies and payment method for TB inpatient care before and after the intervention in Zhenjiang.

Sites	Health Insurance	Baseline (2013)		Health Insurance	New Financing and Payment Policy (2014)	
		Reimbursement Rate of Inpatient Care	Payment Method		Reimbursement Rate of Inpatient Care	Payment Method
Zhenjiang city and Dantu	UEBMI	a. 80%, when cost ≤10,000 CNY b. 90% 10,001–50,000 CNY	Global budget	UEBMI	8000 CNY per case Reimbursement rate: 80%	Case-based payment (the first discharge diagnosis as TB are included)
	URRBMI	a. 30%~40%, when cost ≤10,000 CNY b. 60% 10,001–30,000 CNY c. 70%, ≥30,000 CNY		URRBMI	8000 CNY per case Reimbursement rate: 80%	Case-based payment (the first discharge diagnosis as TB are included)
Yangzhong *	URRBMI	After paying deductible (500 CNY): a. 60% 500–10,000 CNY b. 65% 10,001–50,000 CNY c. 70% 50,001–120,000 CNY	Global budget	URRBMI	8000 CNY per case Reimbursement rate ≥80%	Case-based payment (the first discharge diagnosis as TB are included)
Jurong *	URRBMI	After paying deductible (300–500 CNY): a. 75% at Jurong People’s Hospital b. 65% at other hospitals in Jurong	Global budget	URRBMI	After paying deductible (300–500 CNY): Reimbursement rate ≥80%	Global budget

Data source: local health insurance regulations and reimbursement policies at the county and municipal level before the introduction of new financing mechanism, issued new policy related to financing and payment method during the project period UEBMI: urban employee basic medical insurance; URRBMI: urban and rural residence basic medical insurance. * The reimbursement policy of UEBMI in Yangzhong and Jurong was not retrieved.

**Table 2 ijerph-16-02750-t002:** TB patient characteristics in the pre- and post-intervention surveys in Zhenjiang, in 2013 and 2015, % (*n*).

Category	Subcategory	Pre-Intervention (2013, *n* = 263)	Post-Intervention (2015, *n* = 250)	*p*-Value
Age	17–29	5.7 (15)	5.2 (13)	0.223
	30–59	36.9 (97)	30.0 (75)	
	60+	57.4 (151)	64.8 (162)	
Sex	Male	73.0 (192)	70.8 (177)	0.324
	Female	27.0 (71)	29.2 (73)	
Household income *	Low	22.1 (57)	33.7 (84)	0.005
	Middle	51.9 (134)	48.6 (121)	
	High	25.9 (67)	17.7 (44)	
Health insurance	UEBMI	12.5 (33)	12.8 (32)	0.511
	URRBMI	85.2 (224)	81.1 (202)	0.134
Poverty ^#^	Yes	7.2 (19)	12.0 (30)	0.045
	No	92.8 (244)	88.0 (220)	
Type of patients with TB	New patients	73.4 (193)	72.4 (181)	0.440
	Relapsed patients	26.6 (70)	27.6 (69)	
TB bacteriology diagnosis	Smear Positive	29.0 (72)	34.9 (80)	0.100
	Smear Negative	71.0 (176)	65.1 (149)	
Study counties	Dantu	31.9 (84)	25.2 (63)	0.230
	Yangzhong	31.2 (82)	35.2 (88)	
	Jurong	36.9 (97)	39.6 (99)	

Data source: TB patient survey in 2013 and 2015. * Household income was divided into three groups according to the distributions, high income (top 25), middle income (medium 50) and low income (lowest 25). ^#^ Poverty: Patients living under local poverty line. UEBMI: urban employee basic medical insurance; URRBMI: urban and rural residence basic medical insurance.

**Table 3 ijerph-16-02750-t003:** Hospital admission rate of patients with TB by patient characteristics in Zhenjiang, in 2013 and 2015, % (*n*).

Category	Subcategory	Pre-Intervention (2013, *n* = 263)	Post-Intervention (2015, *n* = 250)	*p*-Value
Total		54.8 (144)	63.6 (159)	0.026
Age	17–29	72.2 (13)	38.5 (5)	0.063
	30–59	50.0 (47)	66.7 (50)	0.029
	60+	55.6 (84)	64.2 (104)	0.123
Sex	Male	54.7 (105)	65.0 (115)	0.044
	Female	54.9 (39)	60.3 (44)	0.520
Household income	Low	49.1 (28)	73.3 (44)	0.001
	Middle	53.4 (55)	67.2 (84)	0.296
	High	59.2 (58)	49.2 (30)	0.189
Health insurance	UEBMI	51.5 (17)	50.0 (16)	0.905
	URRBMI	54.9 (123)	65.8 (133)	0.021
Poverty	Yes	47.4 (9)	63.3 (19)	0.281
	No	55.3 (135)	63.6 (140)	0.069
Type of patients with TB	New patients	56.5 (109)	64.6 (117)	0.107
	Relapsed patients	50.0 (35)	60.9 (42)	0.200
TB bacteriology diagnosis	Smear Positive	68.1 (49)	77.5 (62)	0.054
	Smear Negative	51.1 (90)	55.7 (83)	0.412
Study counties	Dantu	83.3 (70)	84.1 (53)	0.898
	Yangzhong	39.0 (32)	43.2 (38)	0.585
	Jurong	43.3 (42)	68.7 (68)	<0.001

Data source: TB patient survey in 2013 and 2015.

**Table 4 ijerph-16-02750-t004:** Factors associated with hospital admission rates of patients with TB in Zhenjiang, 2015.

Dependent Variable = Hospitalization Status (Yes = 1, No = 0)				
Independent Variables	Odds Ratio	*p*-Value	95% CI	
			Lower	Upper
Age		0.146		
17–29	0.307	0.094	0.077	1.222
30–59	1.241	0.553	0.608	2.533
60+	1	1	1	1
Sex				
Male	1.415	0.324	0.710	2.819
Female	1	1	1	1
Household income		0.083		
Low	3.058	0.029	1.123	8.331
Middle	1.840	0.109	0.872	3.881
High	1	1	1	1
UEBMI				
Yes	0.703	0.559	0.216	2.293
No	1	1	1	1
URRBMI				
Yes	0.802	0.653	0.307	2.095
No	1	1	1	1
Poverty				
Yes	0.440	0.102	0.164	1.178
No	1	1	1	1
Type of patients with TB				
New patients	1.230	0.566	0.606	2.499
Relapsed patients	1	1	1	1
TB bacteriology diagnosis				
Smear Positive	1.918	0.061	0.971	3.789
Smear Negative	1	1	1	1
Study counties		<0.001		
Dantu	3.638	0.006	1.447	9.145
Yangzhong	0.441	0.021	0.220	0.883
Jurong	1	1	1	1

Data source: TB patient survey in 2015.

**Table 5 ijerph-16-02750-t005:** Medical costs of TB inpatient care in designated hospitals in Zhenjiang, in 2013 and 2015 (US dollar *).

Category	Subcategory	Pre-Intervention (2013)		Post-Intervention (2015)		*p*-Value
		Mean	Median	Mean	Median	
Age	17–29	1484.72	582.25	3927.78	4204.80	0.047
	30–59	1968.43	1657.94	3736.49	3152.87	<0.001
	60+	2026.93	1883.09	3156.89	2469.72	0.001
Sex	Male	2004.61	1662.18	3482.31	2938.69	0.045
	Female	1802.62	1467.61	2924.05	2042.74	<0.001
Household income	Low	2246.77	1883.09	3133.51	2938.69	0.166
	Middle	1749.24	1343.45	3412.94	2752.07	<0.001
	High	2041.08	1543.41	3416.15	2671.89	0.029
Health insurance	UEBMI	1396.29	838.03	3911.68	3099.55	<0.001
	URRBMI	2019.02	1714.20	3145.25	2552.09	<0.001
Type of patients with TB	New patients	1845.33	1534.62	3548.41	3085.07	<0.001
	Relapsed patients	2222.69	1838.31	2862.99	2235.32	0.158
TB bacteriology diagnosis	Smear Positive	2073.87	1762.04	4094.76	3577.50	<0.001
	Smear Negative	1816.53	1385.18	2569.61	2350.25	0.003
Study counties	Dantu	1942.49	1657.94	4184.60	3430.81	0.433
	Yangzhong	1889.24	1221.51	2401.84	1800.78	<0.001
	Jurong	1995.38	1689.72	3079.04	2413.05	0.029
Total		1951.80	1658.43	3345.11	2732.00	<0.001

Data source: TB patient survey and medical records of the designated hospitals, a total of 153 cases were successfully matched in Pre-intervention survey, 118 cases were successfully matched in Post-intervention survey. * The medical costs were converted to US dollar using Purchasing Power Parities (PPP) in the study years (OECD National Accounts Statistics: PPP https://data.oecd.org/conversion/purchasing-power-parities-ppp.htm#indicator-chart).

**Table 6 ijerph-16-02750-t006:** Out-of-pocket payment for TB inpatient care in Zhenjiang, in 2013 and 2015, by household income group

Financial Burden	2013 Household Income Group #			2015 Household Income Group #		
	Low (*n* = 26)	Middle (*n* = 73)	High (*n* = 37)	Low (*n* = 22)	Middle (*n* = 78)	High (*n* = 39)
Mean annual household income (US dollar *)	1875.97	11,836.66	29,103.05	458.83	5025.79	19,781.48
Mean out-of-pocket expenditure (US dollar *)	1875.35	1881.92	1676.96	2639.24	4892.78	3168.12
Out-of-pocket expenditure as a percentage of annual household income (%)	99.97	15.89	5.76	575.21	97.35	16.02
% of households that exceed 10% of the household’s annual income	96.15	56.16	18.92	100.00	94.87	51.28
% of households that exceed 20% of the household’s annual income	96.15	27.40	5.41	100.00	85.90	20.51

Data source: TB patient survey in 2013 and 2015, only analyzed the data of TB inpatients, 136 cases in 2013 and 139 cases in 2015. # Household income was divided into three groups according to the distributions, high income (top 25), middle income (medium 50) and low income (lowest 25). * Household income and out-of-pocket expenditure were converted to US dollar using Purchasing Power Parities (PPP) in the study years (OECD National Accounts Statistics: PPP https://data.oecd.org/conversion/purchasing-power-parities-ppp.htm#indicator-chart).

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
