# Peer review of "Factors Influencing Hospitalization Rates and Inpatient Cost of Patients with Tuberculosis in Jiangsu Province, China: An Uncontrolled before and after Study"

_ijerph, 2019, doi:10.3390/ijerph16152750_

Round 1

Reviewer 1 Report

There are minor English and grammatical problems in the text. Please read the paper entirely and correct them. Also, make sure to talk about other methodologies to estimate the cost for providing certain services.

It seems a good paper with only minor changes in the English and also rewriting certain parts such as: abstract, result section ( needs more explanation) and conclusion. In the discussion part , authors should compare their findings with a good number of Economics papers on this topic.

Author Response

Authors would like to sincerely thank you for your time and effort taken to comment on our manuscript. We have carefully addressed each  comments, queries and suggestions. We believe that the manuscript has been greatly improved and hope it has reached your standards.

Please find below a detailed response to your comments.

Point 1: There are minor English and grammatical problems in the text. Please read the paper entirely and correct them. Also, make sure to talk about other methodologies to estimate the cost for providing certain services.

Response 1: We have invited a native English speaking co-author to check the language.

Point2:It seems a good paper with only minor changes in the  English and also rewriting certain parts such as: abstract, result section ( needs more explanation) and conclusion. In the discussion part , authors should compare  their findings with a good number of Economics papers on this topic.

Response 2: We have modified the abstract and rest of the manuscript. 

Reviewer 2 Report

Review for the International Journal of Environmental Research and Public Health

Regarding the manuscript entitled ‘Factors influencing hospitalization rates and impatient cost of patients with tuberculosis….’ , I offer the following reaction.

The paper is interesting and I think the topic is important. Assessing tuberculosis rates, profile of patients, and hospitalization ‘before and after’ a significant change in financing policy is a good focus. The results of this study would be particularly relevant to government planners intent on reducing financial burdens and hopefully improving the health care of particularly the least affluent.  The authors have secured good datasets and the overall methodological approach seems logical. The literature review is brief (perhaps too brief), but the studies cited are relevant.  There are some problems, particularly with clarity in the methodology, but a few other issues need to be addressed as well. I will elaborate below.

1) While the paper is readable, there are a number of grammar errors and inconsistencies.  Subsequent drafts should address these issues.

2) In the method section, section 2.1.2 is unclear.  It is stated that the sample is based on the first 90 patients.  Then, at the end of the paragraph, 263 and 250 patients are included in the analysis.  I do not see the connection here.

3) Three townships/streets in each county were selected based on distance (far, not far and close).  There needs to be more clarity here.  There is a difference between a township and a street (at least in my experience); I don’t think these should be used interchangeably. The choice of far, not far and close is vague.  Why is this important?  Should actual distances by given?

3) The first paragraph in the Results section and Table 1 are not really results.  This is background information on the change in financial policy.  This section should be in the methodology or perhaps in the introduction.

4) CNY needs to be initialized, I assume this is Chinese Yuan Renminbi (currency) but not everyone will know this.

5) What statistical test are the p values on Tables 2 and 3 based on?  There are a number of p references made on the text as well.  Only logistic regression is mentioned, but all statistics tests should be revealed in the method section.

6) I don’t see how the first sentence in section 3.2 can be correct or at least it is not clear. 54.8% is not higher than 63.6%.

7) In general, there are a lot of tables but not much description.  Perhaps a couple more sentences highlighting key results from the tables should be added to the text.

8) The first sentence of the third paragraph in the Discussion seems like a contradiction.  It is stated that new financing failed to regulate hospital admission rates, but hospital admission rates did go down over this time period.  So, if rates went down, then the new plan was successful in this regard, right?

9) Overall, I think the Discussion section is good, but more description (which may be partially speculative) as to why the new financing approach was not successful (as the authors seem to imply) could be added.  It would be interesting to get more information about the hospital profit-driven motives and lack of monitoring that are so problematic as explanations. Also, are the failures of the system also linked to an inability of officials to properly understand the profile of typical TB patients?  In other words, the discussion could be linked more with the results from Tables 2, 3 and 5 in its critique of financial policy reforms failure.

Author Response

Authors would like to sincerely thank you for your time and effort taken to comment on our manuscript. We have carefully addressed each  comments, queries and suggestions. We believe that the manuscript has been greatly improved and hope it has reached your standards.

Please find below a detailed response to your comments:

Point 1: The paper is interesting and I think the topic is important. Assessing tuberculosis rates, profile of patients, and hospitalization ‘before and after’ a significant change in financing policy is a good focus. The results of this study would be particularly relevant to government planners intent on reducing financial burdens and hopefully improving the health care of particularly the least affluent.  The authors have secured good datasets and the overall methodological approach seems logical. The literature review is brief (perhaps too brief), but the studies cited are relevant.  There are some problems, particularly with clarity in the methodology, but a few other issues need to be addressed as well. I will elaborate below.

Response 1: We responded to the comments. Please see below.

Point 2: While the paper is readable, there are a number of grammar errors and inconsistencies.  Subsequent drafts should address these issues.

Response 2: We have invited a native English speaking co-author to check the language.

Point 3: In the method section, section 2.1.2 is unclear.  It is stated that the sample is based on the first 90 patients.  Then, at the end of the paragraph, 263 and 250 patients are included in the analysis.  I do not see the connection here.

Response 3: Small number of sampled patients refused to participate in the survey. Hence, 263 (out of 270 sampled patients) patients in 2013 and 250 cases in 2015 completed the survey and included in the analysis. We have clarified in the text (page 4, line 63).

Point 4: Three townships/streets in each county were selected based on distance (far, not far and close).  There needs to be more clarity here.  There is a difference between a township and a street (at least in my experience); I don’t think these should be used interchangeably. The choice of far, not far and close is vague.  Why is this important?  Should actual distances by given?

Response 4: We considered how convenient for patients to access to TB care, and thus selected townships (not street, deleted in the text, page 4, line 55) based on the distance to the county TB designated hospital (far, not far and close according to local TB care providers’ perceptions).

Point 5: The first paragraph in the Results section and Table 1 are not really results.  This is background information on the change in financial policy.  This section should be in the methodology or perhaps in the introduction.

Response 5: Thanks for the suggestion. This part was based on policy review. We summarized health insurance reimbursement policy and payment methods before and after the implementation project. We considered it as part of results and keep it in the result section.    

Point 6: CNY needs to be initialized, I assume this is Chinese Yuan Renminbi (currency) but not everyone will know this.

Response 6: Thanks for the suggestion. We have added in page 5, line 128.

Point 7: What statistical test are the p values on Tables 2 and 3 based on?  There are a number of p references made on the text as well.  Only logistic regression is mentioned, but all statistics tests should be revealed in the method section.

Response 7: Chi-square tests was used to test differences. We have added in page 5, line 99.

Point 8: I don’t see how the first sentence in section 3.2 can be correct or at least it is not clear. 54.8% is not higher than 63.6%.

Response 8: We have reformulated the sentence (page 10).

Point 9: In general, there are a lot of tables but not much description.  Perhaps a couple more sentences highlighting key results from the tables should be added to the text.

Response 9: We have summarized the key findings from tables.   

Point 10: The first sentence of the third paragraph in the Discussion seems like a contradiction.  It is stated that new financing failed to regulate hospital admission rates, but hospital admission rates did go down over this time period.  So, if rates went down, then the new plan was successful in this regard, right?

Response 10: The hospital admission rate was higher after the implementation of the new financing policy. We also studied number of hospital admission per patient. It was slightly lower in 2015 (1.16 per patient) than in 2013 (1.26 per patient). We revised the sentence in page 16, line 61-62.

Point 11: Overall, I think the Discussion section is good, but more description (which may be partially speculative) as to why the new financing approach was not successful (as the authors seem to imply) could be added.  It would be interesting to get more information about the hospital profit-driven motives and lack of monitoring that are so problematic as explanations. Also, are the failures of the system also linked to an inability of officials to properly understand the profile of typical TB patients?  In other words, the discussion could be linked more with the results from Tables 2, 3 and 5 in its critique of financial policy reforms failure.

Response 11: We agree. We did qualitative study to understand factors impacting on the policy implementation, but it is out of scope of this paper. The research team has submitted another manuscript on the lessons learned from the policy implementation, and that one is under review.
